# A Funnel Type PVDF Underwater Energy Harvester with Spiral Structure Mounted on the Harvester Support

**DOI:** 10.3390/mi13040579

**Published:** 2022-04-07

**Authors:** Jongkil Lee, Jinhyo Ahn, Hyundu Jin, Chong Hyun Lee, Yoonsang Jeong, Kibae Lee, Hee-Seon Seo, Yohan Cho

**Affiliations:** 1Department of Mechanical Engineering Education, Andong National University, Andong 36729, Korea; jlee@anu.ac.kr (J.L.); malong8484@naver.com (J.A.); comando5@naver.com (H.J.); 2Department of Ocean System Engineering, Jeju National University, Jeju 63243, Korea; jyssolt@naver.com (Y.J.); kibae0211@gmail.com (K.L.); 3Agency for Defense Development, Daejeon 34186, Korea; hsseo@add.re.kr (H.-S.S.); yhcho@add.re.kr (Y.C.)

**Keywords:** funnel type energy harvester, cantilever type PVDF, flow-induced vibration, voltage doubler rectifier, full bridge rectifier

## Abstract

For the purpose of stably supplying electric power to the underwater wireless sensor, the energy harvesting technology in which a voltage is obtained by generating displacement in a piezoelectric material using flow-induced vibration is one of the most attractive research fields. The funnel type energy harvester (FTEH) with PVDF proposed in this study is an energy harvester in which the inlet has a larger cross-sectional area than the outlet and a spiral structure is inserted to generate a vortex flow at the inlet. Based on numerical analysis, when PVDF with L = 100 mm and t = 1 mm was used, the electric power of 39 μW was generated at flow velocity of 0.25 m/s. In experiment the average RMS voltage of FTEH increased by 0.0209 V when the flow velocity increased by 1 m/s. When measured at 0.25 m/s flow velocity for 25 s, it was shown that voltage doubler rectifier (VDR) generated a voltage of 133.4 mV, 2.25 times larger than that of full bridge rectifier (FBR), and the energy charged in the capacitor was 44.3 nJ, 14% higher in VDR than that of the FBR. In addition, the VDR can deliver power of 17.75 μW for 1 kΩ load. It is shown that if the voltage generated by the FTEH using the flow velocity is stored using the VDR electric circuit, it will greatly contribute to the stable power supply of the underwater wireless sensor.

## 1. Introduction

It is true that energy harvesting technology based on vibration to drive various miniaturized and low-power sensors has attracted a lot of attention for many years. Especially among the different types of energy sources available in water, vibrational energy is known to be the most attractive because it is a kinetic energy that is abundant, readily accessible, and can be easily converted into electrical energy using piezoelectric, electromagnetic or electrostatic principles [1,2]. There are several piezoelectric materials that can convert this vibrational energy in water into electrical energy and are used in a variety of sensing and actuation applications. In energy harvesting, technologies using piezoelectricity, thermoelectricity, and triboelectricity are being actively studied. Among them, piezoelectric energy harvesting (PEH) is widely used by many researchers [3,4,5,6,7,8] because it has advantages of high power density and various application fields compared to other technologies.

PEH is based on the phenomenon of generating a current flow by creating a potential difference through mechanical energy and vibrational displacement using an element with a piezoelectric effect. The most used piezoelectric material for PEH is PZT, which has excellent cost-effectiveness and mass productivity, but is very weak to impact due to the characteristic of ceramics. In terms of durability of materials, polyvinylidene fluoride (PVDF), fiber-type macro fiber composite (MFC), PMN-PT, PMN-PZT, etc. are being actively studied [9]. In underwater energy harvesting using piezoelectric materials, important power generating devices can be classified into two main categories. As in the study of Erturk et al. [8] and Shan et al. [10], those using cantilevers and those using flow around circular rods or cables [11].

Erturk et al. [8] manufactured a cantilever type underwater energy harvesting device in the form of a caudal fin using MFC of the concept of piezohydroelasticity, which is capable of underwater propulsion and energy harvesting, and verified the energy harvesting performance through experiments. It was predicted that 2.5 mAh of power could be charged through about 20 h of charging for vibration with a 0.5 g acceleration of 56 Hz. Recently, Shan et al. [10] verified the performance of an MFC-based energy harvesting device using an underwater vortex environment. They built a mathematical model for an energy harvesting device using a piezoelectric material in a vortex environment and predicted the energy generation performance. The performance of the energy harvesting device according to the flow rate were also conducted. It was stated that a maximum of 1.32 μW of energy could be generated when a vortex was generated with a 30 mm diameter cylinder at a flow rate of 0.5 m/s.

Bezanson et al. [12] reported that supply utilizing vortex induced vibration energy (SURVIVE) is a structure in which a thin cantilever is attached to the surface of a cymbal-type piezoelectric transducer and these are installed on the electronics housing so that the vortex flow generated by the fluid flow vibrates the cantilever. The design targets ocean current of 0.25 m/s and each generator was found to generate a minimum of 6 mW based on the experimental result.

Taylor et al. [13] developed a flag device composed of two embedded PVDF layers by applying the body and movement of an eel. The vortex alternates, which causes the flag to flutter and consequently generates electricity from the piezoelectric material due to charge separation [14].

As a study using a piezoelectric cantilever beam, Akaydin et al. [7] measured the energy harvesting ability of PVDF beams in unstable turbulence (Reynolds number > 10,000). In the wake of the turbulent flow of a circular cylinder, the fluid passes along the surface of the beam placed at an optimized position, and the beam is placed on a vibrating turbulent boundary layer to generate power.

Power generated by using PEH basically is alternating current (AC) because it is based on deformation caused by vibration. Since AC current cannot be used directly in batteries and direct current (DC) power applications, DC conversion through a rectifier circuit is required. The basic rectifier circuit used in PEH is full bridge rectifier (FBR) [2,15,16]. When the FBR composed of 4 diodes is used, forward voltage drop of diode in a low voltage circuit is a loss that cannot be ignored. To overcome this shortcoming, a circuit adopting a voltage doubler rectifier (VDR) has been proposed [17]. The VDR has two advantages over the FBR; (1) because of half usage of diodes, voltage drop is small.; and (2), the voltage output of PEH can be increased up to 2 times [18].

As a method for generating flow induced vibration in underwater, there can be a method using a vortex flow generated on the rear surface of an object with a circular cross section, and a method using a vortex flow generated on the surface of an object having a cantilever shape. However, the cantilever type, which has a relatively wider surface area than a circular cross-section and vibrates sensitively to small changes in external force, was selected in this paper as a structure that can self-excited the residual vibration in water for a long time.

In this study, a cantilever type funnel type energy harvester (FTEH) using PVDF, a piezoelectric material, was fabricated. PVDF is considered as a material that can vibrate freely according to the fluid flow and can obtain a large amount of vibration displacement rather than other rigid PZT material. PVDF film is relatively a simple monomer structure. It is made of organic polymer and is resistant to corrosion in underwater. FTEH has a spiral screw shape mounted on the support and the inlet is wide and the outlet is narrow. Vibration displacement generation according to the design parameters of FTEH was analyzed through numerical simulation. The effectiveness of the FTEH was verified by manufacturing an experimental device and installing VDR and FBR at the output terminal of the FTEH to measure the amount of power generated according to the flow rate through the experiment.

## 2. Numerical Simulations and Results

In the energy harvester, the piezoelectric energy is generated from the displacement of the PVDF piezoelectric body installed in the funnel-shaped outlet part. $w_{rel}\left(x,t\right)$ is the vibration response, i.e., transverse displacement at position *x* and time *t*, $v\left(t\right)$ is the voltage response across the external resistive load $R_{l}$. Based on the standard modal analysis procedure the vibration response is expressed in terms of the modal mechanical coordinate that gives the transverse vibration displacement $y_{r}\left(t\right)$ and the mode shapes $\varphi_{r}\left(x\right)$ as [1]
(1)$$w_{rel}\left(x,t\right)=\overset{\infty }{\underset{r=1}{\sum }}\varphi_{r}\left(x\right)y_{r}\left(t\right)$$

The electromechanically coupled ordinary differential equations in modal coordinates are [1]
(2)$$\frac{d^{2}y_{r}\left(t\right)}{dt^{2}}+2\delta_{r}\omega_{r}\frac{dy_{r}\left(t\right)}{dt}+\omega_{r}^{2}y_{r}\left(t\right)-\theta_{r}v\left(t\right)=f_{r}\left(t\right)$$
(3)$$\tilde{C}\frac{dv\left(t\right)}{dt}+\frac{v\left(t\right)}{R_{l}}+\overset{\infty }{\underset{r=1}{\sum }}\theta_{r}\frac{dy_{r}\left(t\right)}{dt}=0$$
where $\omega_{r}$ is the undamped natural frequency in constant electric field conditions, $\delta_{r}$ is the modal mechanical damping ratio, $f_{r}\left(t\right)$ is a modal forcing function, $\theta_{r}$ is the modal electromechanical coupling, and $\tilde{C}$ is a permittivity component of the piezo-ceramic layers. Hence, these Equations (1)−(3) can predict the coupled system dynamics and one obtains voltage response which depends on the vibration displacement [1].

Numerical analysis and optimization studies of energy harvester devices using flow-induced vibrations around FTEH generated by the fluid flow in water were conducted. A displacement occurred in the PVDF piezoelectric film due to the pressure change due to the flow of the fluid, and a bidirectional coupling analysis was performed in which this displacement again affects the flow field. The material properties of the PVDF were 1780 kg/m^3^ in density, 2500 MPa in Young’s modulus, and 0.35 of Poisson’s ratio. For the fluid 11,952 of Reynolds number was applied to the depth of 10 m.

### 2.1. Energy Harvester Model with FTEH

As for the model used for the analysis, an energy harvester device with a funnel-type inlet shape was devised as shown in Figure 1. The flow of the fluid runs from the left with a wide inlet to the right with a narrow inlet. At the end of the outlet, a PVDF piezoelectric body for harvesting energy using vortex vibration caused by the flow of fluid is assembled.

In the funnel-type structure shown in Figure 1, the cross-sectional area at the end of the energy harvester is smaller than the inlet, so an increase in flow rate can be expected according to Bernoulli’s theorem. In general, it would be a good approach to perform the optimal design of the energy harvester according to the shape of the FTEH. However, in this study, the shape and size (cross-sectional area of inlet and outlet) of the FTEH had to be analyzed in a condition where it was specially limited for the purpose of use. Therefore, in this study, due to the limited purpose of use in the marine environment, the amount of vibration displacement was observed with respect to changes in the length and thickness of PVDF and the rate of inflow of ocean currents.

Assuming that only a piezoelectric body without a fluid collecting device is independently placed in water and an energy harvester model with a funnel-type inlet, set to case (a) and case (b), respectively, and calculate the velocity distribution, pressure distribution, and vibration displacement as shown in Figure 2 and Figure 3. Figure 2a and Figure 3a show the velocity distribution and pressure distribution in the shape without a funnel, respectively. Figure 2b and Figure 3b respectively show the velocity distribution and pressure distribution in the FTEH shape. In this case the input flow velocity was set to 0.24 m/s, which is the average current velocity. As a result, the maximum speed at the outlet of case (b), where the funnel type fluid collector exists, increased about 1.9 times from 0.24 m/s to 0.45 m/s. Accordingly, the pressure difference was also relatively large. As shown in Figure 2, it can be seen that FTEH generates a lot of vortex flow in the velocity distribution, which increases the amount of vibration in PVDF.

Figure 4 shows the vibrational displacement at the tip of the PVDF piezoelectric body. Unlike case (a), which is a simple piezoelectric model with little vibration displacement, in case (b), a funnel-type energy harvester model, the maximum displacement is 0.01 mm, which can be seen to generate a much larger vibration displacement compared to case (a).

Figure 4c shows the structure in which the shape of the funnel is symmetrical on the inlet side and the outlet side, and the vibrational displacement is shown in (d) when the fluid flows under the same conditions. As can be seen from Figure 4a,b,d, it can be found that the shape as shown in Figure 1b had a lot of vibrational displacement.

### 2.2. Optimal Design of FTEH

In the FTEH model, which has a larger vibration displacement than the simple piezoelectric model without an fluid collecting device, the vibration displacement according to the thickness, length, and input flow rate of the piezoelectric element was measured to determine the trend. A PVDF piezoelectric film was used as the piezoelectric material, and the thickness (t) of the film-shaped piezoelectric material was decreased from 1.0 mm to 0.75 mm at 0.125 mm decrements, and analysis was performed in three cases, respectively. The film length (L) was increased from 50 mm to 100 mm in 25 mm increments, and the inlet velocity (V∞) was increased or decreased based on the average seawater velocity of 0.10 m/s, 0.24 m/s, and 0.50 m/s. A total of three cases were determined and analysis was performed. The design variables are summarized in Figure 5 and Table 1. The analysis results are graphically shown in Figure 6, Figure 7, Figure 8 and Figure 9. When comparing the vibration displacement according to the thickness and length of the piezoelectric body, the inlet speed was fixed at 0.24 m/s, which is the average seawater speed.

Figure 6 and Figure 7 show the vibration displacement and frequency spectrum for each inlet velocity of the funnel-type energy harvester device, respectively. As the inlet speed increased from 0.1 m/s to 0.5 m/s, the vibration displacement increased. As the speed increased, the high frequency vibration displacement was measured on the PVDF piezoelectric film. In Figure 8, the vibration displacement according to the length of the PVDF piezoelectric body is compared when the average current velocity V∞ = 0.24 m/s. As the length of the piezoelectric body increased from 50 to 100 mm, the vibration displacement increased. As shown in Table 2, in the case of the PVDF piezoelectric film, as the length increased by 50%, the vibration displacement increased by 2.3 times and 100% was increased by 8.6 times based on the maximum displacement difference.

In Figure 9, the vibration displacement according to the thickness of the PVDF piezoelectric body is compared when the average current velocity V∞ = 0.24 m/s. As the thickness of the piezoelectric material decreased from 1.0 mm to 0.75 mm, the vibrational displacement increased. As shown in Table 3, in the case of PVDF piezoelectric film, as the thickness was decreased by 12.5%, the vibrational displacement increased by 4.2 times and 25% was increased by 5.7 times.

The predicted electric energy harvesting value of the FTEH device derived from the optimal design analysis is 2.299~43.17 μW as shown in Figure 10. When PVDF with a length of 100 mm and a thickness of 1 mm was used, it was found that power of 39 μW was generated at a flow velocity of 0.25 m/s. Therefore, as shown in Figure 10, as the vibration displacement of PVDF increases, the generated power is proportionally higher, indicating that the vibration displacement and the generated voltage have a proportional relationship.

## 3. Power and Energy Generated by FTEH

The proposed PVDF harvester generates an irregular low voltage signal according to the underwater flow and the signal cannot be used for battery charging and DC power application. Therefore, a full-wave rectification circuit capable of voltage boosting is required. In this paper, we use a VDR as a rectifier circuit and present its improved performance over typical FBR.

The VDR and FBR connected to the PVDF element are shown in Figure 11. In Figure 11. $v_{i}$ is the input voltage, $v_{o}$ is the output voltage, D is rectifying diodes, and C is rectifying capacitor. The PVDF harvester is represented as an equivalent model composed of internal capacitance $C_{s}\;$ and source voltage $\;v_{s}$ [19].

Suppose that voltage source $v_{s}$ a sinusoidal wave having different frequency at every half cycle and $w_{n}$ is the angular frequency at nth cycle as shown in Figure 12.

Then the maximum average output power of the VDR $\langle P_{VD}{\left(t_{n}\right)\rangle }_{max}$ and FBR $\langle P_{FB}{\left(t_{n}\right)\rangle }_{max}$ during half cycle [$t_{n-1}<t<t_{n}$] are as follows [18]
(4)$$\langle P_{VD}{\left(t_{n}\right)\rangle }_{max}=\frac{C_{s}w_{n}}{\pi }{\left(V_{OC\left(n\right)}-V_{D}\right)}^{2}$$
(5)$$\langle P_{FB}{\left(t_{n}\right)\rangle }_{max}=\frac{C_{s}w_{n}}{\pi }{\left(V_{OC\left(n\right)}-2V_{D}\right)}^{2}$$
where $V_{D}$ is the voltage drop due to diode D, $V_{oc\left(n\right)}$ is the peak voltage of the nth half cycle. Note that the only one $V_{D}$ occurs in the VDR whereas the FBR has two $V_{D}$ in the path of the current. The accumulated energy via N frequency signals of VDR $E_{VD}\left(t_{N}\right)$ and FBR $E_{FB}\left(t_{N}\right)$ can be written as
(6)$$E_{VD}\left(t_{N}\right)=\int_{t_{0}}^{t_{N}}\langle P_{VD}{\left(t\right)\rangle }_{max}dt$$
(7)$$E_{FB}\left(t_{N}\right)=\int_{t_{0}}^{t_{N}}\langle P_{FB}{\left(t\right)\rangle }_{max}dt$$

Since direct measurement of the source voltage of the actual harvester is not possible, it is necessary to estimate the source voltage from the output observed by the measuring equipment.

To estimate the source voltage $v_{s}$, the open circuit voltage of the harvester was measured using an oscilloscope as shown in Figure 13. In Figure 13a, $R_{osc}$ is the resistor 10 MΩ of the oscilloscope, $v_{osc}$ is open-circuit voltage of harvester measured by oscilloscope, $C_{s}$ is internal capacitance 10.26 nF of the PVDF. Figure 13b shows the $v_{osc}$ of the harvester actually measured in water. The circuit in Figure 13a can be interpreted as a first-order analog high-pass filter with one resistance and capacitance so that reverse transfer function of the circuit $H_{inv}\left(s\right)\;$ can be obtained as follows:(8)$$H_{inv}\left(s\right)=\frac{V_{s}\left(s\right)}{V_{osc}\left(s\right)}=\frac{1+s\tau }{s\tau }\;\;$$
where $\tau =R_{osc}C_{s}$ is the time constant. Finally, source voltage of the harvester $v_{s}\left(t\right)\;$ can be obtained as follows:(9)$$v_{s}\left(t\right)=\int_{0}^{t}v_{osc}\left(x\right)\left[\delta \left(t-x\right)+\frac{1}{\tau }\right]dx.$$

Figure 14a shows source voltage $v_{s}\;$ of the harvester estimated using Equation (6) from the measured open-circuit voltage $v_{osc}$ in Figure 13b.

To evaluate the performance of the VDR, we estimate voltage source by approximating as a sinusoidal signal having different angular frequencies in five time intervals as shown in red dotted line in Figure 14a.

When the approximated sinusoidal signal is used and $V_{D}$ = 0.09, the output energy of each rectifier calculated using Equations (3) and (4) for 100 s are shown in Figure 14b. As shown in Figure 14, the VDR generates 17% more energy than the FBR because of smaller voltage drops.

By using the estimated harvester source voltage in Figure 14a, we made a SPICE model via the same configuration shown in Figure 11, such as $C_{s}$ = 10.26 nF, C = 22 μF and BAT43 as diode model D. Figure 15 shows the simulation results of the output voltage and charging energy of each rectifier for 100 s.

As shown in Figure 15, the VDR shows 2.12 times higher voltage and 13% more charged energy than those of the FBR at 100 s when the voltage converges. These results confirm that the VDR is advantageous for a low-voltage voltage source compared to the FBR.

## 4. Experimental Result and Discussions

### 4.1. Energy Harvester Model with FTEH

As shown in Figure 16, the fluid circulation device consists of an upper water tank and a lower water tank. The upper water tank has a standard of (300 mm × 300 mm × 1000 mm), and the four sides except for the bottom surface are made of transparent acrylic to facilitate observation, and the bottom surface is made of white opaque acrylic. The upper water tank is fixed on the aluminum frame and placed above the lower water tank. The standard of the lower water tank is (300 mm × 330 mm × 1400 mm), and the storage capacity is about 140 L. Wheels are attached to the four corners of the lower end of the aluminum frame to facilitate movement when measuring water intake, drainage, and experiments.

A flow meter was installed at the inlet of the upper water tank to measure the flow rate of the fluid during the experiment, and the flow rate was controlled by a control valve installed at the front end of the flow meter. The fluid transmitted by the pump passes through the energy harvester device in a stable state through a mesh-shaped flow distributor installed at the inlet of the upper water tank. In addition, the upper water tank outlet was designed to allow the fluid to overflow as shown in Figure 16 to adjust the water level. The outlet surface was designed in a structure that could be opened and closed through several holes to control the flow of fluid. The pump used in the experiment is an underwater pump for drainage, and the pumping amount is 185 L/min.

After repeatedly measuring and marking the distance corresponding to the average speed of the fluid required for the experiment with a flow meter, FTEH was installed at the marked location. The flow meter used in the experiment with a resolution of 0.01 m/s, and the average speed of 30 s was repeatedly measured and used. The flow meter installed in the upper water tank is shown in Figure 17a. As illustrated in Figure 17b, the bottom surface of the upper water tank was designed in a rail shape to accurately move and fix the position of support supporting the energy harvester device.

### 4.2. Fabrication of FTEH

Based on theoretical analysis and case study results, funnel-type energy harvester prototypes of various initial models were produced. A soft type PVDF was used as the piezoelectric material, a bidirectional model, as shown in Figure 17, which can be used in both vertical and horizontal directions on the water surface, was used as the final model of the screw type harvester device, which increases the vortex of the fluid passing through FTEH.

### 4.3. Experimental Results and Discussions

Harvester voltage data were measured at three fluid speed of 0.25 m/s, 0.5 m/s, and 1 m/s. The open voltage of the harvester and the voltage of the rectifying capacitor are measured with an Agilent E4085 oscilloscope.

We measured RMS open-circuit voltages at three flow speeds for 100 s with and without funnel installation. As shown in Figure 18b, the proposed FTEH harvester generates 2.38 times higher RMS voltage on average than normal energy harvester. The dotted line in Figure 18b is the 1st order fit of FTEH’s RMS voltage according to the speed and can be expressed as following:(10)$$V_{FTEH}=0.0209u+0.1127,$$
where u is the flow speed. Note that at every 1 m/s of flow speed increase leads to 0.0209 V increase. Using Equation (7), the power and energy according to the flow rate are calculated as follows:(11)$$P_{FTEH}={\left(0.0209u+0.1127\right)}^{2}/R_{L},$$
(12)$$E_{FTEH}={\left(0.0209u+0.1127\right)}^{2}t/R_{L},$$
where $R_{L}$ is the load resistance, and t is measurement time. If $R_{L}=10\;M\Omega$, t=100 s, u=0.25m/s, then energy of FTEH becomes 139 nJ.

The energy was calculated by measuring the voltage charged for 25 s with flow speed of 0.25 m/s. Figure 19 shows the voltage and energy at the rectifying capacitor. As shown in Figure 19a,b, the VDR can generate 2.25 times higher voltage and 14% more charged energy than those of the FBR at 25 s. These results confirm that the VDR is advantageous for a low-voltage voltage source compared to the FBR. In addition, Figure 19c shows the output power according to charging time for 1 kΩ load. As shown in Figure 19c, the VDR can deliver power of 17.75 μW
for 25 s.

As shown in Figure 19a, the proposed FTEH and VDR generate voltage of 133.4 mV for 25 s. The proposed VDR circuit, however can generate voltage and power enough to drive underwater temperature sensor by connecting multiple FTEHs and VDR circuits in series as described in Appendix A.

## 5. Conclusions

A method of supplying power by applying energy harvesting technology to a wireless sensor used underwater is a very useful technology. A method of obtaining voltage by generating vibration displacement in a piezoelectric material using flow induced vibration generated by the flow of an underwater ocean current is a technology that can stably supply power to a wireless sensor.

In this study, FTEH using flow-induced vibration in underwater was devised and its usefulness was verified through numerical analysis and experiments. FTEH is a funnel type in which the fluid inlet has a larger cross-sectional area than the outlet, and PVDF is installed at the outlet to generate voltage. At the inlet side, a spiral structure was mounted on the FTEH’s support to generate a vortex flow of the fluid. As a result of the numerical analysis, it was found that the structure with the funnel generated more vibration displacement than the structure without the funnel, and it was confirmed that as the flow rate increased, the thickness of the PVDF decreased, and the length of the PVDF increased. When PVDF with a length of 100 mm and a thickness of 1 mm was used, it was found that power of 39 μW was generated at a flow velocity of 0.25 m/s. In the energy storage circuit development, it was confirmed that the VDR stores 13% more energy than the FBR.

An experimental device equipped with FTEH was manufactured, and the electric power generated by FTEH was stored in the rectify circuit while the flow velocity was changed from 0.25 m/s to 1.0 m/s. As a result of the experiment, it was confirmed that the average RMS voltage of FTEH increased by 0.0209 V when the flow rate increased by 1 m/s. In order to see the performance of the electric circuit, the voltage charged in the rectifying capacitor of each rectifier was measured and the power and energy were compared. When measured for 25 s at a flow rate of 0.25 m/s, it was confirmed that VDR has a voltage 2.25 times greater than FBR. The energy charged in the capacitor was measured, and it was confirmed that the VDR was charged as much as 44.3 nJ, which is 14% higher than the FBR. In the future, if the effective voltage generation in the water of FTEH is stored using the VDR electric circuit, it is judged that it will greatly contribute to the stable power supply of the wireless sensor.

## Figures and Tables

**Figure 1 micromachines-13-00579-f001:**
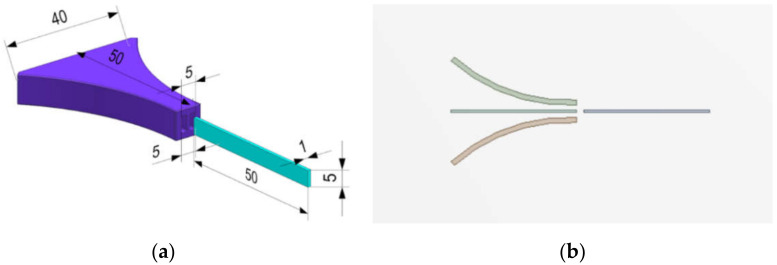
Modeling of the FTEH: (**a**) 3D CAD Model; (**b**) 2.5D Model.

**Figure 2 micromachines-13-00579-f002:**
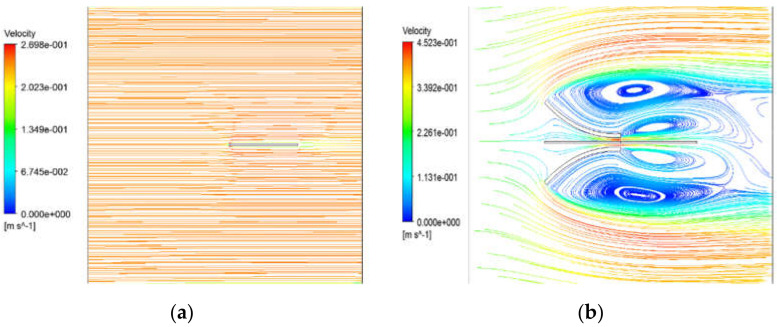
Velocity distribution by numerical simulation: (**a**) without FTEH; (**b**) with FTEH.

**Figure 3 micromachines-13-00579-f003:**
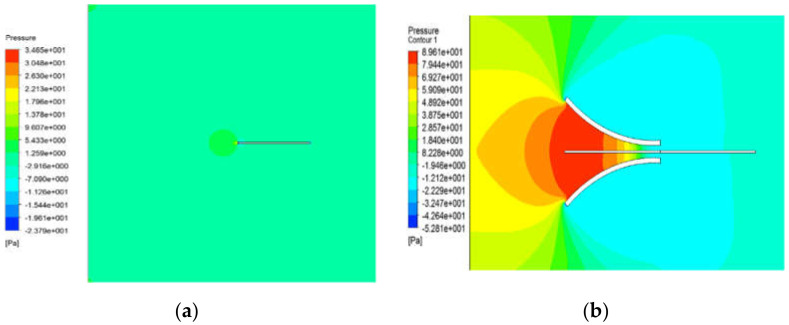
Pressure distribution by numerical simulation: (**a**) without FTEH; (**b**) with FTEH.

**Figure 4 micromachines-13-00579-f004:**
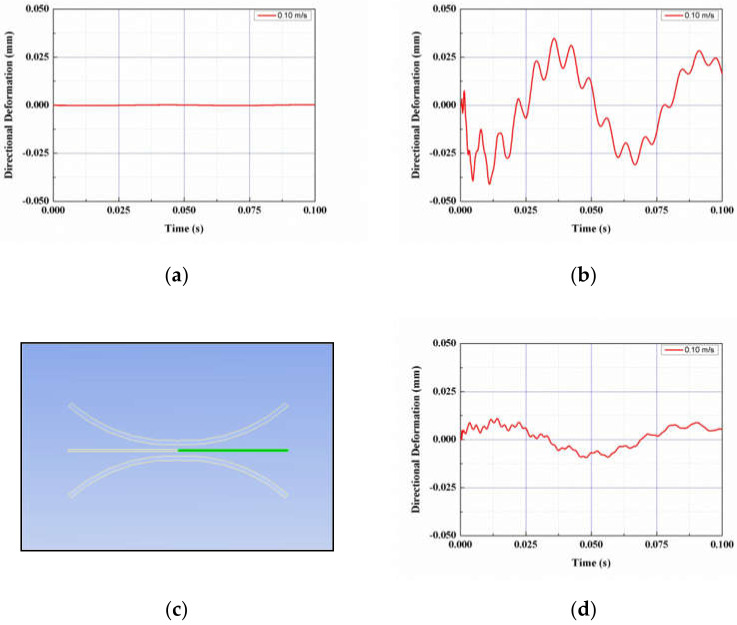
Vibration displacement by numerical simulation: (**a**) without FTEH; (**b**) with FTEH; (**c**) symmetric shape of the FETH; (**d**) vibration displacement of the case (**c**).

**Figure 5 micromachines-13-00579-f005:**
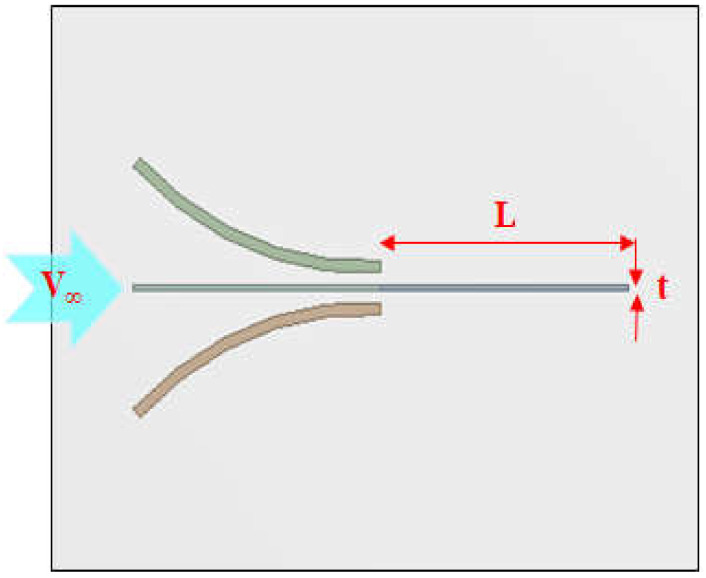
Schematic diagram of the FTEH and dimensions.

**Figure 6 micromachines-13-00579-f006:**
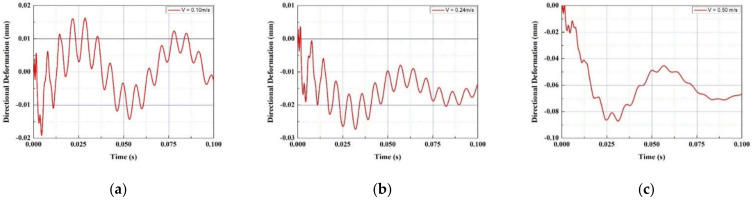
Vibration displacement with inlet velocity (V∞): (**a**) 0.10 m/s; (**b**) 0.24 m/s (**c**) 0.50 m/s.

**Figure 7 micromachines-13-00579-f007:**
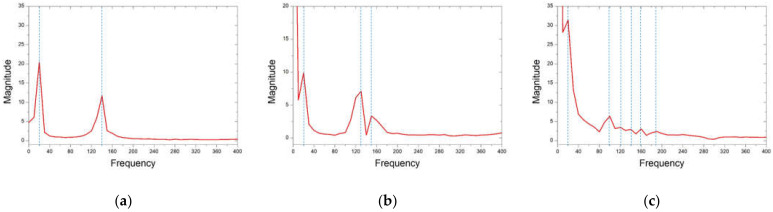
Vibration frequency spectrum under the variation of inlet velocity (V∞): (**a**) 0.10 m/s; (**b**) 0.24 m/s; (**c**) 0.50 m/s.

**Figure 8 micromachines-13-00579-f008:**
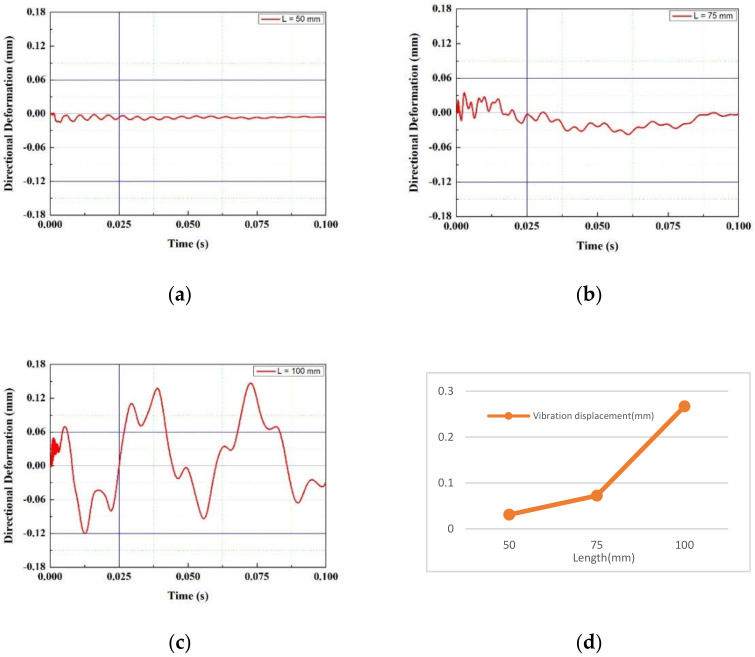
Vibration displacement along piezoelectric body length (L): (**a**) 50 mm; (**b**) 75 mm; (**c**) 100 mm; (**d**) 50~100 mm.

**Figure 9 micromachines-13-00579-f009:**
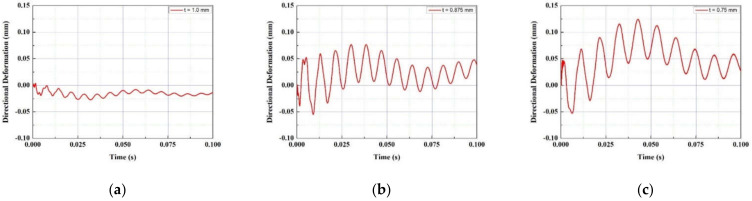
Vibration displacement according to piezoelectric thickness (t): (**a**) 1.0 mm; (**b**) 0.875 mm; (**c**) 0.75 mm.

**Figure 10 micromachines-13-00579-f010:**
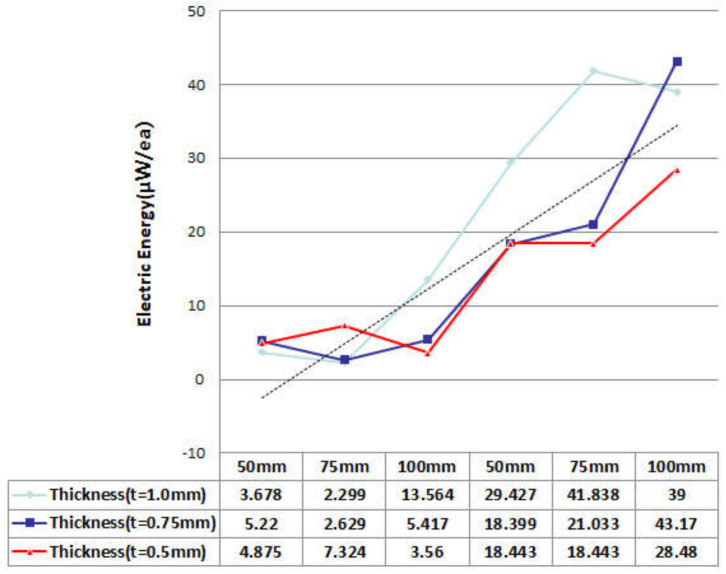
Power generation estimation (μW/ea).

**Figure 11 micromachines-13-00579-f011:**
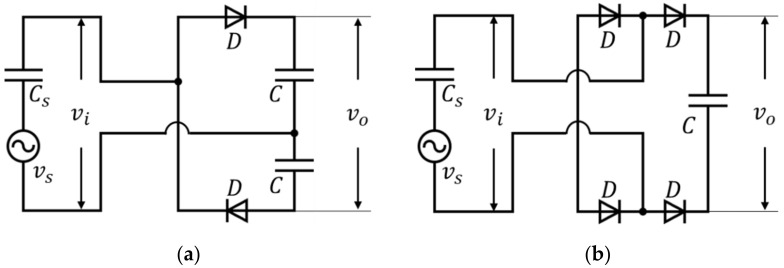
Rectifier circuits: (**a**) VDR; (**b**) FBR.

**Figure 12 micromachines-13-00579-f012:**
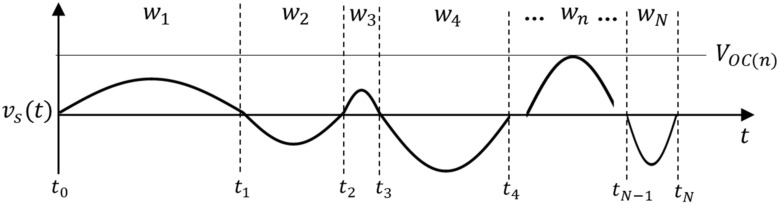
Sinusoidal voltage source model.

**Figure 13 micromachines-13-00579-f013:**
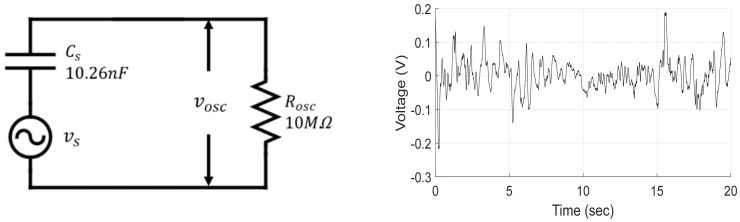
(**a**) Open voltage measurement system; (**b**) Measured voltage.

**Figure 14 micromachines-13-00579-f014:**
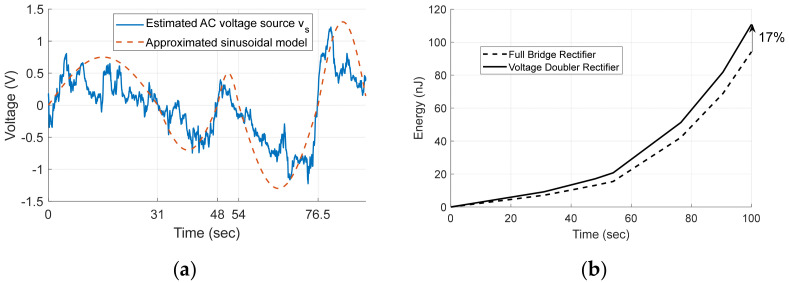
(**a**) Estimated voltage source; (**b**) Estimated output energy.

**Figure 15 micromachines-13-00579-f015:**
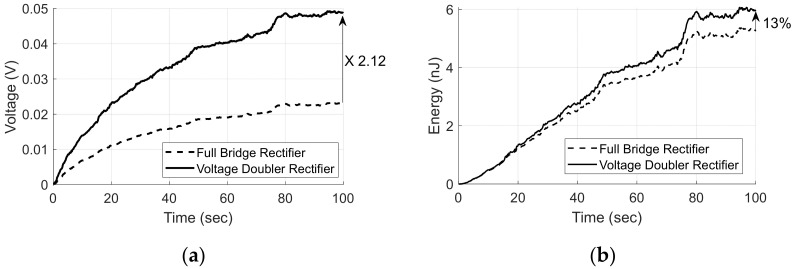
VDR and FBR outputs (**a**) Rectifier voltage; (**b**) Charged energy.

**Figure 16 micromachines-13-00579-f016:**
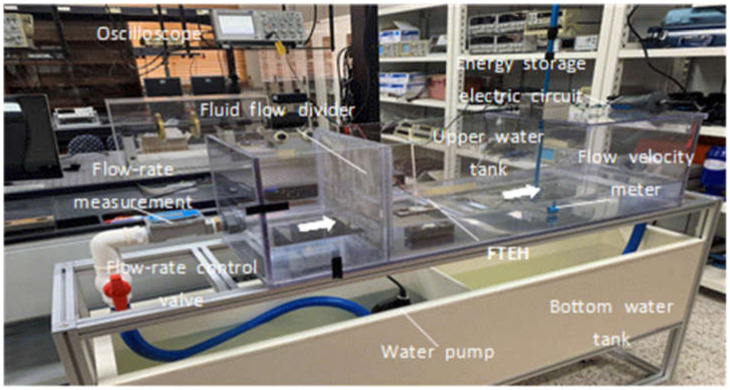
Experimental set up.

**Figure 17 micromachines-13-00579-f017:**
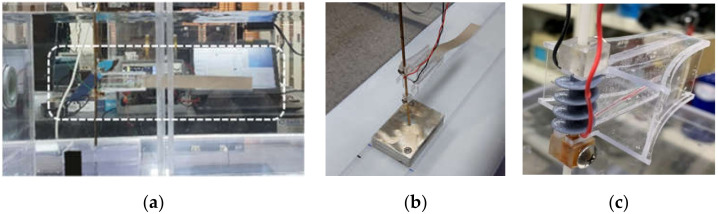
FTEH and experimental set up: (**a**) side view of the FTEH; (**b**) FTEH fixed in rail shape; (**c**) FTEH with spiral structure mounted on the harvester support.

**Figure 18 micromachines-13-00579-f018:**
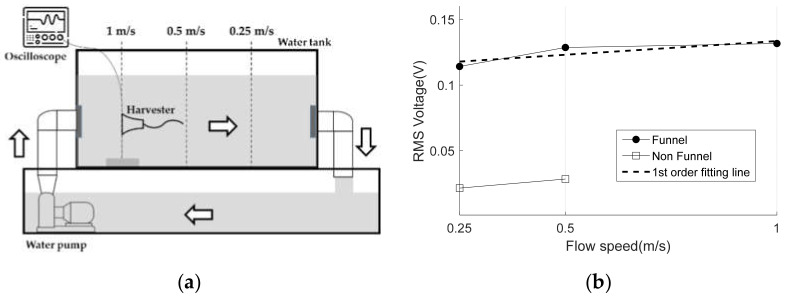
(**a**) Schematics of measurement setup; (**b**) RMS voltages according to flow speed.

**Figure 19 micromachines-13-00579-f019:**
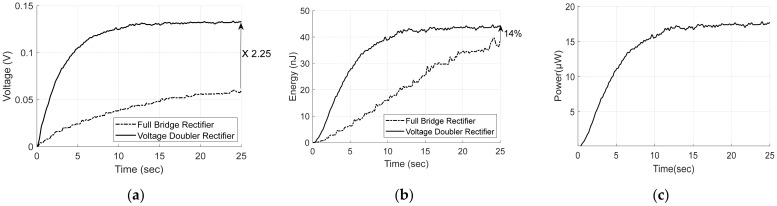
VDR and FBR outputs (**a**) Rectifier voltage; (**b**) Charged energy; (**c**) Output power of VDR for 1 kΩ load.

**Table 1 micromachines-13-00579-t001:** FTEH design parameters.

Material	Film Thickness, t	Film Length, L	Inlet Velocity, V_∞_
PVDF	1.0 mm	50 mm	0.10 m/s
	0.875 mm	75 mm	0.24 m/s
	0.75 mm	100 mm	0.50 m/s

**Table 2 micromachines-13-00579-t002:** Vibration displacement along PVDF piezoelectric body length.

Length	L = 50 mm	L = 7 5 mm	L = 100 mm
Max.	0.003723	0.034846	0.14703
Min.	−0.027319	−0.037583	−0.12009
maximum displacement difference	0.031042	0.072429	0.26712
rate of increase	1	2.33325817	8.60511565

**Table 3 micromachines-13-00579-t003:** Vibration displacement according to PVDF piezoelectric material thickness.

Thickness	t = 1.0 mm	t = 0.875 mm	t = 0.75 mm
Max.	0.003723	0.076791	0.12458
Min.	−0.027319	−0.055017	−0.05272
maximum displacement difference	0.031042	0.131808	0.1773
rate of increase	1	4.24611816	5.71161652

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
