# Peer review of "A Funnel Type PVDF Underwater Energy Harvester with Spiral Structure Mounted on the Harvester Support"

_micromachines, 2022, doi:10.3390/mi13040579_

Round 1
Reviewer 1 Report
In this manuscript, the authors reported a PVDF based funnel type energy harvester (FTEH) for underwater energy collection. Thanks to the funnel structure, which has a larger inlet and a smaller outlet, the flow rate is increased. The vibration displacement generation based on different design parameters of PVDF was analyzed through numerical simulation. The performance of the device with voltage double rectifier (VDR) and full bridge rectifier (FBR) were tested and discussed and has shown that VDR electric circuit will contribute to the stable power supply of the wireless sensor. Although the idea of funnel shaped piezoelectric generator is interesting, more studies are expected to carry out such as more optimization analysis of the sensor design, analysis of the real test, etc. What’s more, the scope of Micromachines is about the micro/nano-scaled structures, materials, devices and system. The current work doesn't contain enough micro/nano study. As a result, I think this work is proper for publication in Micromachines.
Major comment:
1. In the introduction section, it’s better to clarify more about the reason for choosing PVDF instead of PZT and other fiber-type composites to highlight the advantage of this material in this application.
2. According to the authors, there are two basic categories of piezoelectric generators for underwater energy harvesting, one is cantilevers and the other is based on the flow around circular rods or cables. Can you please explain the reason why you choose the first design strategy?
3. The funnel shape can influence the flow speed and pressure. The analysis of the funnel design should be added.
4. The authors only studied the displacement at the tip but the relationship between the voltage and the displacement is not clear. Please explain how the displacement is related to the output voltage in both simulation and the experiments.
Author Response
We appreciate reviewing time in reviewing our paper and providing valuable comments. The authors have carefully considered the comments and tried our best to address every one of them. We hope the revised manuscript meet your high standards.
In attached file, we provide the point-by point responses. All modifications in the manuscript have been highlighted in red.

Reviewer 2 Report
In this paper, the author proposed a funnel type PVDF underwater energy harvester, which has a high efficiency. The authors carried out simulations, rectifying circuits and optimization. Thank you for the authors contribution to the energy harvesting area. The following are several my concern on the paper. Please check.
1 In the abstract, the voltage and power should be given to show the research clear.
2 Figure 4 is not clear. A vector image may be a good choice. Check out the whole paper.
3 The voltage is very low, can the authors show me an underwater sensor, which works in so low a voltage level.
4 Further, the power is not provided. As you know, energy(nJ level) is not the most important factor, but the power is the one which determined whether the low power sensor can work continuously.
5 the voltage is very low, can the VDR voltage works? Please provide an experiment.
6 Figure 7 and Figure 8, I suggest I would be not wise to the provide the Deformation along time. It is not visual. The deformation along L itself may be better.
Author Response

(The authors gave the same response as above.)

Round 2
Reviewer 1 Report
Although the authors answered some of my questions (comment #1 and #2), there are still several issues to be addressed.
In my comment #3, my concern is how the shape of the funnel (e.g. the length, the size of the input and output, etc.) influences the flow speed and pressure rather than if the structure influences the results. This is an important topic to study to optimize the structure of the whole device. However, I don' see the update related to this problem.
In my comment #4, I would like to see some formulas to get the relationship between the displacement and generated voltage. The voltage is the target in this work, however, the authors only studied the tip displacement in the simulation. The relation between the displacement and voltage is not clear. What's more, the structure with different thicknesses should have different curvatures during the vibration. As a result, the max displacement is not enough to represent the whole deformation from the piezoelectric material. The authors only give the conclusion that the two parameters have a proportional relationship, which is not sufficient.
Author Response
We appreciate reviewing time in reviewing our paper and providing valuable comments. The authors have carefully considered the comments and tried our best to address every one of them. We hope the revised manuscript meet your high standards.

Reviewer 2 Report
I Suggest accept in present form.
An advice, 100 harvesting units are not easy to be controlled. Therefore, I suggest more consideration on this point in the future.
Author Response
We appreciate reviewing time in reviewing our paper and providing valuable comments. Also, we appreciate your decision on accepting our manuscript in present form.
Round 3
Reviewer 1 Report
The authors answered my questions. I don't have further concerns.